# Phosphorylation-Dependent SERS Readout for Activity Assay of Protein Kinase A in Cell Extracts

**DOI:** 10.3390/nano10030575

**Published:** 2020-03-22

**Authors:** Renyong Liu, Chenggen Xie, Yehan Yan, Lin Hu, Suhua Wang, Khalid A. Alamry, Hadi M. Marwani, Lijuan Chen

**Affiliations:** 1Key Laboratory of Biomimetic Sensor and Detecting Technology of Anhui Province, School of Materials and Chemical Engineering, West Anhui University, Lu’an 237012, Anhui, China; ahlry@mail.ustc.edu.cn (R.L.); cgxie@wxc.edu.cn (C.X.); yanyh@mail.ustc.edu.cn (Y.Y.); 2Anhui Province Key Laboratory of Condensed Matter Physics at Extreme Conditions, High Magnetic Field Laboratory, Chinese Academy of Sciences, Hefei 230031, Anhui, China; 3College of Environmental Science and Engineering, North China Electirc Power University, Bejing 102205, China; wangsuhua@ncepu.edu.cn; 4Chemistry Department, Faculty of Science, King Abdulaziz University, Jeddah 21589, Saudi Arabia; k_alamry@yahoo.com (K.A.A.); hmarwani@kau.edu.sa (H.M.M.)

**Keywords:** phosphorylation-dependent SERS readout, spermine-functionalized AgNPs, hot spots, PKA activity, cell extracts

## Abstract

Protein kinases are key regulators of cell function, the abnormal activity of which may induce several human diseases, including cancers. Therefore, it is of great significance to develop a sensitive and reliable method for assaying protein kinase activities in real biological samples. Here, we report the phosphorylation-dependent surface-enhanced Raman scattering (SERS) readout of spermine-functionalized silver nanoparticles (AgNPs) for protein kinase A (PKA) activity assay in cell extracts. In this assay, the presence of PKA would phosphorylate and alter the net charge states of Raman dye-labeled substrate peptides, and the resulting anionic products could absorb onto the AgNPs with cationic surface charge through electrostatic attraction. Meanwhile, the Raman signals of dyes labeled on peptides were strongly enhanced by the aggregated AgNPs with interparticle hot spots formed in assay buffer. The SERS readout was directly proportional to the PKA activity in a wide range of 0.0001–0.5 U·μL^−1^ with a detection limit as low as 0.00003 U·μL^−1^. Moreover, the proposed SERS-based assay for the PKA activity was successfully applied to monitoring the activity and inhibition of PKA in real biological samples, particularly in cell extracts, which would be beneficial for kinase-related disease diagnostics and inhibitor screening.

## 1. Introduction

Protein kinases are a class of enzymes that catalyze the transfer of phosphate groups from nucleotide triphosphates (often ATP) to specific residues of proteins, thereby inducing conformational changes of target proteins and affecting their functions. The post-translational modification plays an important role in regulating intracellular processes, such as metabolism [1], signal transduction [2,3,4], and DNA replication and repair [5]. The abnormal activity of protein kinases may induce several human diseases [6,7,8]. Especially, it has been found that there are lots of highly sensitive and specific phosphorylated proteins or peptides in the body fluids or tissues from cancer patients, indicating that phosphorylation by the specific kinase is closely related to the occurrence and development of cancers [9,10,11,12].

Various assay systems, including mass spectrometry [13], radioactivity [14], immunoprecipitation [15], fluorescence [16,17,18,19], colorimetry [20], and electrochemistry [21,22,23] based methods, have been developed for the detection of protein kinases and the corresponding phosphorylated substrates. Although these methods can be selected for assessing kinase activities according to the changes of the structures and physicochemical properties of proteins or peptides before and after modification, they still have some drawbacks. The mass spectrometry detection always needs to separate and enrich the extremely low amount of phosphorylated substrates in samples. Radioactive methods inevitably use harmful radioactive substances. Immunoprecipitation assays are dependent on the expensive phosphorylated peptide antibodies with poor specificity. As one of the most important biomarkers, the levels of protein kinases are extremely low in real biological samples. Therefore, there is an urgent need to develop a simple, rapid, low cost, and highly sensitive method for assaying the activity of protein kinases and their phosphorylated substrates.

With the integration of high sensitivity, a unique fingerprint feature, resistance to water interference, and photo-bleaching abilities, the surface-enhanced Raman scattering (SERS) technique has become one of the most attractive approaches for biological detection and imaging [24,25,26,27,28,29]. In the last decades, many efforts have been devoted to producing the intense electromagnetic field essential for the Raman enhancement by adjusting the surface plasmon property of metallic nanostructures [30,31,32]. It is commonly regarded that highly SERS-active metal nanoparticles in their aggregates can produce interparticle hot spots and achieve an enormous Raman enhancement [24,33,34,35,36]. However, the aggregated nanostructures still face a challenge in reliable and quantitative measurements of targets, due to the inconsistency of nanoparticle aggregation [37], and ineffective contact with targets [38].

Herein, we report a phosphorylation-dependent SERS readout strategy for protein kinase A (PKA) activity assay using spermine-functionalized silver nanoparticles (AgNPs). Phosphorylation by PKA would alter the net charge state of the specific substrate labeled with Raman dye 5-carboxytetramethylrhodamine (here referred to as TAMRA-LRRASLG), and the resulting anionic products could absorb onto aggregated AgNPs with a cationic surface charge through electrostatic attraction in assay buffer. Thus, the signals of TAMRA labeled on peptides would be remarkably enhanced due to the close proximity of interparticle hot spots. Herein, we showed how the positively charged AgNPs were particularly suitable as SERS platforms for PKA activity assay. It is expected that the dependence of phosphorylation on SERS readout can help reduce background signals originated from the directly dye-labeled nanoparticles [27] and avoid false positive results, which is favorable for the sensitive and reliable assay of the activity and inhibition of PKA in cell extracts. Considering recent progress in the synthesis and cellular uptake of functionalized nanoparticles [39,40], and studies on protein–nanoparticles interaction [41], there will be a strong driving force to expand the SERS strategy to real-time monitoring of the specific kinase activity in living cells in future.

## 2. Materials and Methods

### 2.1. Materials

Spermine tetrahydrochloride, silver nitrate (AgNO_3_), sodium borohydride (NaBH_4_), adenosine 5′-triphosphate disodium salt hydrate (ATP), protein kinase A (PKA, from bovine heart), and dithiothreitol (DTT) were purchased from Sigma–Aldrich (Shanghai, China). Dihydrochloride hydrate (H-89) was obtained from J&K Scientific Ltd. (Shanghai, China). Protein kinase B (PKB) was purchased from Bio-Techne China Co., Ltd. (Shanghai, China). Forskolin, 3-isobutyl-1-methylxanthine (IBMX), and Tris-HCl buffer solution (pH 7.5) were purchased from Sangon Biotech (Shanghai, China). Substrate peptide for PKA (TAMRA-LRRASLG, 98%), and phosphorylated substrate peptide were purchased from GL Biochem (Shanghai, China). Magnesium chloride (MgCl_2_), sodium chloride (NaCl), ethylenediaminetetraacetic acid (EDTA), glycerol, and dimethyl sulfoxide (DMSO) were obtained from Shanghai Chemicals Ltd. (Shanghai, China). The human cervical carcinoma cells (HeLa) were purchased from the Type Culture Collection of the Chinese Academy of Sciences (Shanghai, China). All other reagents were of A.R. grade and used as received without further purification. Ultrapure water with a resistivity of 18.2 MΩ·cm was used.

### 2.2. Preparation of Spermine-Functionalized AgNPs

The used AgNPs were synthesized in an aqueous solution by reducing AgNO_3_ with NaBH_4_ in the presence of spermine tetrahydrochloride [42]. Briefly, 10 mL of 1 mM aqueous AgNO_3_ and 5 μL of 0.1 M spermine tetrahydrochloride were sequently added into a three-neck flask, and the mixture was degassed with nitrogen for 30 min. Then, 25 μL of a freshly prepared 0.1 M aqueous NaBH_4_ solution was added quickly, and stirred vigorously for 20 min. The obtained solution was directly used for the following SERS measurements unless otherwise noted.

### 2.3. Procedure for PKA Activity Assay and Inhibition Study

PKA standard solutions were prepared by adding different amounts of PKA into the storing solution (20 mM Tris-HCl buffer, 50 mM NaCl, 1 mM EDTA, 2 mM DTT, and 50% glycerol). For each assay, 5 μL of PKA standard solutions at various concentrations (0~20 U·μL^−1^) were mixed with the reaction solution (70 μL of 50 mM Tris-HCl buffer, 10 μL of 10 μM TAMRA-LRRASLG, 5 μL of 50 μM ATP, 10 μL of 5 mM MgCl_2_), allowed to react for 1 h at 37 °C in dark, and diluted with 300 μL of 50 mM Tris-HCl buffer. Then 100 μL spermine-AgNPs solution was added to the 100 μL diluted reaction solution. After they were incubated for 15 min, the Raman spectra were recorded. For the PKA inhibition study, the experiments were conducted under the above-mentioned conditions except that the fixed concentration of PKA (1 U·μL^−1^) was pretreated with different concentrations of H-89 (0~2 μM), respectively. Test data were measured using three assay replicates for each sample (*n* = 3).

### 2.4. SERS Detection of PKA Activity in Cell Lysate

HeLa cells were cultured in Dulbecco’s modification of Eagle’s medium (DMEM) supplemented with 10% fetal bovine serum (FBS) at 37 °C under 5% CO_2_. The cultured cells were lysed with lysis buffer and centrifugated (12000 rpm, 10 min). The supernates were diluted as the lysates (10 μg·mL^−1^) and the PKA activity in cell lysates was assayed according to the protocol described above. For stimulation experiments, the culture medium was replaced by a serum-free medium, and different concentrations of forskolin/IBMX (10 μM/20 μM and 25 μM/50 μM in DMSO) were respectively added into the medium for activating the intracellular PKA. As a control, equal volumes of DMSO were added into the medium for comparison. In addition, the corresponding inhibition experiments of PKA in cell lysates were performed by pretreatment with 5 μM H-89.

### 2.5. Characterization

The morphology of AgNPs was examined by transmission electron microscopy (TEM, JEOL 2010, Tokyo, Japan) operated at 200 kV accelerating voltage. UV-vis absorption spectra of AgNPs and their aggregates were recorded with a Shimadzu UV-2550 spectrophotometer (Shimadzu, Tokyo, Japan). Zeta potential measurements of AgNPs were carried out using a Nano ZS ZEN3600 Zetasizer (Malvern Instruments Ltd., Malvern, England). Raman spectra were recorded using the DXR confocal microscopy Raman system (Thermo Fisher Scientific Inc., Waltham, MA, USA) with a 532-nm excitation laser and 10× objective lens. To minimize the changes induced by the laser power and obtain moderately enhanced signals, the SERS spectra were recorded in solution with a relatively high but safe laser power of 10 mW, and the accumulation time was 10 s.

## 3. Results and Discussion

Figure 1a illustrates the principle of the detection system for PKA activity assay based on the phosphorylation-dependent SERS readout strategy. The spermine-functionalized AgNPs with positive surface charges (more accurately their interparticle aggregation formed by the electrostatic attraction between AgNPs and anions in the assay buffer) were used here as enhancing SERS substrates. The TAMRA-labeled substrate peptides with arginine residues possessed the positive charge, while TAMRA-LRRASLG phosphorylated by PKA had the negative charge due to the introduction of phosphate backbone. Thus, the resulting anionic phosphorylated products were able to adsorb onto aggregated AgNPs with a cationic surface charge through electrostatic attraction. As dyes labeled on peptides were positioned in the interparticle hot spots and close to the surface of AgNPs, the huge enhancement of TAMRA signals were able to be recorded. In the absence of phosphorylation, no Raman signal was able to be obtained for substrate peptides TAMRA-LRRASLG due to the ineffective contact with AgNPs.

The positively charged AgNPs (+ 43.2 mV) employed here were synthesized in the presence of spermine tetrahydrochloride, which has been demonstrated as an effective ligand for modifying AgNPs and improving SERS sensing performance of DNA [42]. TEM examinations showed that the synthesized AgNPs had an average particle size of ~27 nm and were highly monodispersive in aqueous solution (Appendix A), whereas the particle aggregation occurred either in Tris-HCl buffer (Appendix A), or in the reaction solution after phosphorylation (Figure 1b). The corresponding aggregated states of AgNPs were also monitored by a UV-vis spectroscopy (Appendix A). When AgNPs were added into the Tris-HCl buffer, the original absorbance of AgNPs at 392 nm disappeared and a broad absorption band (centered at ~500 nm) from their aggregates appeared. The almost identical result was observed after AgNPs were added in PKA reaction solution, indicating that the existence of both TAMRA-LRRASLG and phosphorylated products had little influence on the aggregation of AgNPs. Importantly, no obvious nanoparticle precipitation occurred over a time period of 60 min. We envisioned that the aggregated nanostructures with interparticle hot spots were favorable for producing high and stable SERS signals. It should be noted that the spermine-functionalized AgNPs showed a low background signal (Figure 1c, curve 1). To demonstrate the feasibility of the proposed method, SERS spectra of the mixtures of AgNPs and TAMRA-LRRASLG under different conditions were recorded. It was shown that only extremely weak or negligible Raman intensities were detected in the absence of PKA (curve 2) or ATP (curve 3), while the dominant peaks of TAMRA (1364, 1510, 1536 and 1650 cm^−1^) with remarkably enhanced Raman intensities [43] were recorded from the mixtures of AgNPs and TAMRA-LRRASLG after the phosphorylation reaction with PKA and ATP (curve 4). These results suggest that the phosphorylation-dependent SERS readout strategy is particularly suitable for PKA activity assay. 

Considering that the excessive free ATP may compete with phosphorylated TAMRA-LRRASLG for AgNPs, the effect of ATP concentration on SERS readout was carefully investigated (Figure 2a). When the concentration of ATP was higher than 2.5 μM, a significant decrease of SERS signal intensity was observed. The same results were obtained by the use of the synthetic phosphorylated products instead of TAMRA-LRRASLG phosphorylated by PKA (data not shown). These indicated that PKA was saturated with the employment of 2.5 μM ATP and excess of such concentration would affect the assaying performance due to the competition binding to AgNPs between free ATP and phosphorylated products, which is consistent with the results reported by fluorescence [44]. To obtain enhanced signals, and to avoid the negative effect of free ATP on phosphorylated peptides/AgNPs interaction, 2.5 μM ATP was employed in the following assay unless otherwise noted. 

Figure 2b showed the results of the PKA activity in the assay system under the optimized experimental conditions. As is evident, the SERS signals of TAMRA labeled on peptides were enhanced with the increasing concentration of PKA, demonstrating that the increased amounts of phosphorylated TAMRA-LRRASLG were produced, and subsequently entered the conjunctions among the aggregated AgNPs. Notably, the SERS signal intensities reached a plateau when the concentration of PKA was higher than 0.5 U·μL^−1^. The relative intensity (I_PKA_/I_blank_) of the strongest peak at 1650 cm^−1^ (assigning to the symmetric in-place C–H bend of the xanthene ring [43]) was used for the quantitative evaluation of PKA activity, and exhibited a very wide linear relationship with the PKA concentration ranging from 0.0001 U·μL^−1^ to 0.5 U·μL^−1^ (*R*^2^ = 0.9949, Figure 2c). The limit of detection for PKA was calculated as low as 0.00003 U·μL^−1^ based on three standard deviations above the value of the blank sample. To the best of our knowledge, the limit of detection was higher than the recently reported PKA assay [17,18,19], clearly demonstrating the advantage of high sensitivity of SERS technique. Moreover, the excellent specificity of this assay system was also verified by replacing PKA with PKB, the Raman intensity for which was only comparable to that of the blank sample (Figure 2d).

Moreover, the potential application of this method was tested in the inhibition assay by the use of H-89, a specific PKA inhibitor, with different concentrations. As seen in Figure 3, the signal intensity indeed decreased as the concentrations of H-89 increased. The results indicated that the PKA activity could be effectively inhibited by H-89, leading to the reduced amounts of phosphorylated TAMRA-LRRASLG situated at the conjunctions of the aggregated AgNPs and the declined SERS signal intensity. Therefore, the proposed method has great potential for applications in screening and for the quantitative detection of inhibitors. 

The sensitivity and applicability of this system were further demonstrated by assaying the activity and inhibition of PKA in real biological samples particularly in cell extracts. HeLa cells, taking for example, were first treated by different concentrations of forskolin and 3-isobutyl-1-methylxanthine (IBMX), which have been demonstrated as effective at stimulating drugs for the activation of cAMP-dependent PKA [45,46]. The effect of various ions and salts existing in cell extracts on the aggregation of AgNPs was first examined by UV-vis spectroscopy. The characteristic absorption band of aggregated AgNPs was still able to be observed and the corresponding intensity had little or no change after AgNPs were added in the cell lysate solution and incubated for up to 60 min (Appendix A). That is to say, no obvious nanoparticle precipitation occurred in the assay solution. Then, the activity and inhibition of PKA in cell extracts were monitored by the phosphorylation-dependent SERS assay. Interestingly, even without drug stimulation, the distinguishable SERS signals from the extracts were able to be detected due to the presence of PKA in HeLa cells (Appendix A), which has been demonstrated by electrophoresis [47] and fluorescence [18]. Meanwhile, the drugs stimulated cell lysates all aroused obviously enhanced Raman signals, which gradually increased with the concentration of forskolin and IBMX (Figure 4a). Notably, the SERS signal intensity at 1650 cm^−1^ reached a plateau when the incubation time was longer than 15 min and the stable SERS intensities kept almost identical for up to 60 min, suggesting the good stability of spermine-functionalized AgNPs over long periods of time. Moreover, all of these signal enhancements were effectively suppressed by H-89 (Figure 4b). These results indicate that the phosphorylation-dependent SERS assay is feasible and reliable for in vitro assessment of PKA activity and its variation in real biological samples.

## 4. Conclusions

In summary, we have developed a novel assay system for assessing the activity of PKA based on the spermine-functionalized AgNPs and their effect on the SERS of dye-labeled peptides. While the use of positively charged AgNPs ensures electrostatic attraction between TAMRA-LRRASLG phosphorylated by PKA and the enhancing substrates with interparticle hot spots, the PKA activities can be sensitively and reliably assayed in cell extracts by the SERS readout of TAMRA. In particular, the phosphorylation-dependent SERS readout strategy can effectively avoid false positive signals caused from commonly used dye-labeled nanoparticle-based systems. Since protein kinases are potentially important biomarkers [9,48,49], the proposed method is expected to exhibit exciting prospects in clinical diagnosis and drug discovery.

## Figures and Tables

**Figure 1 nanomaterials-10-00575-f001:**
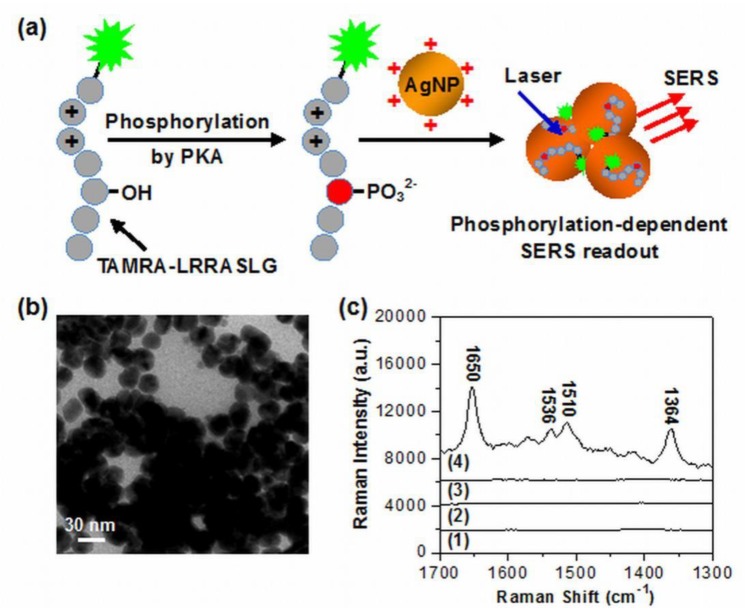
(**a**) Schematic illustration for PKA activity assay based on the phosphorylation-dependent surface-enhanced Raman scattering (SERS) readout. (**b**) TEM image of the aggregated silver nanoparticles (AgNPs) in the reaction solution after phosphorylation. (**c**) SERS spectra of (1) pure spermine-functionalized AgNPs, the mixtures of spermine-functionalized AgNPs and 1 μM substrate peptides labeled with Raman dye 5-carboxytetramethylrhodamine (TAMRA-LRRASLG) after the phosphorylation reaction with (2) only 2.5 μM ATP, (3) only 1 U·μL^−1^ PKA, and (4) 2.5 μM ATP and 1 U·μL^−1^ PKA (spectra were off set).

**Figure 2 nanomaterials-10-00575-f002:**
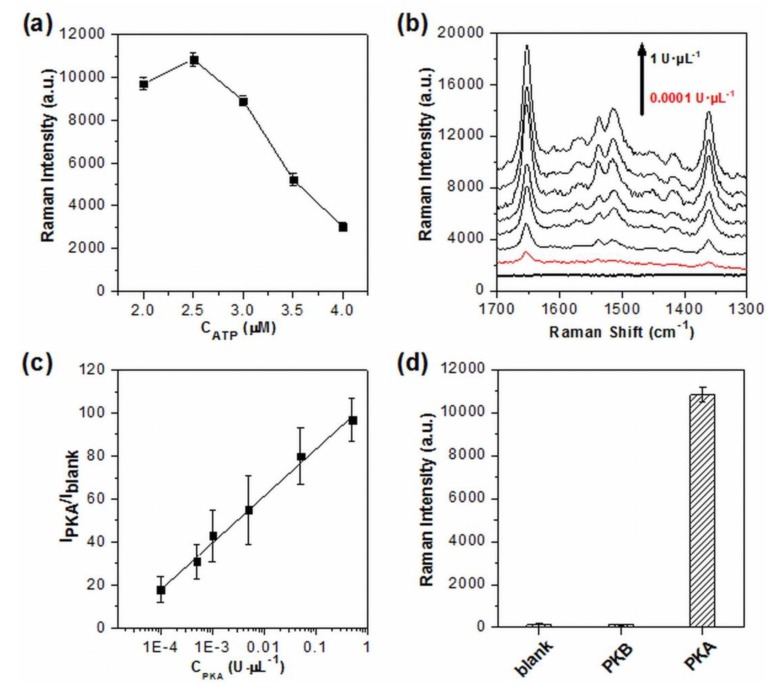
(**a**) Comparison of SERS signal intensity at 1650 cm^−1^ as a function of the concentrations of ATP (2, 2.5, 3, 3.5 and 4 μM, experimental conditions: TAMRA-LRRASLG, 1 μM; PKA, 1 U·μL^−1^). (**b**) SERS spectra obtained from the assay system with different concentrations of PKA (0.0001, 0.0005, 0.001, 0.005, 0.05, 0.5 and 1 U·μL^−1^, spectra were off set and the bottom dark line represents the blank). (**c**) Linear correlation of the relative signal intensity (I_PKA_/I_blank_) versus PKA concentration (I_PKA_ and I_blank_ represent the signal intensity at 1650 cm^−1^ in the presence and absence of PKA, respectively). (**d**) The selectivity of the SERS assay system to PKA over PKB (experimental conditions: PKA, 1 U·μL^−1^; PKB, 1 U·μL^−1^; TAMRA-LRRASLG, 1 μM; ATP, 2.5 μM). The intensity of 1650 cm^−1^ was used for the evaluation of SERS readout, and error bars represent one standard deviation of three replicates (*n* = 3).

**Figure 3 nanomaterials-10-00575-f003:**
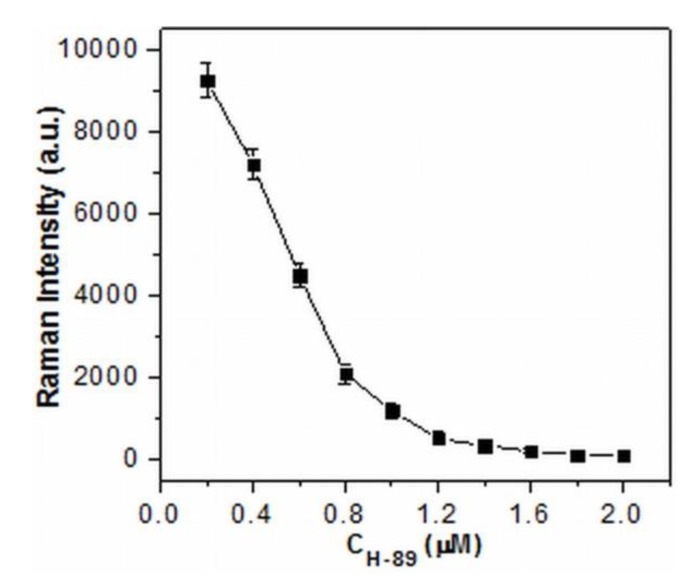
SERS intensity of 1650 cm^−1^ as a function of the concentration of PKA inhibitor H-89 (experimental conditions: TAMRA-LRRASLG, 1 μM; PKA, 1.0 U·μL^−1^; H-89, 0-2 μM).

**Figure 4 nanomaterials-10-00575-f004:**
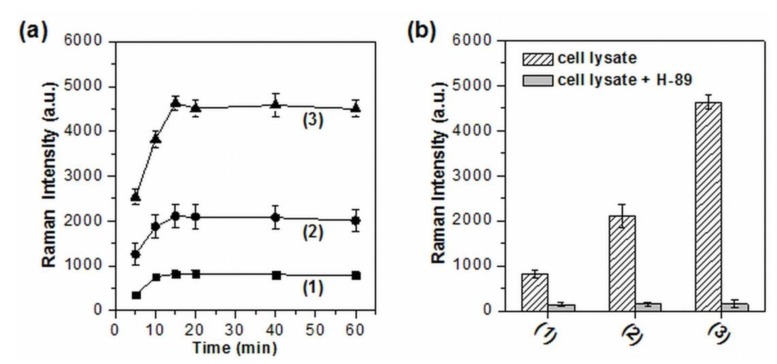
(**a**) The evolution of the SERS signals from the HeLa cell extracts with increasing incubation time. (**b**) The inhibition assay of PKA in HeLa cell extracts. Curve 1–3 represent the cell extracts without stimulation, with the stimulation of 10 μM forskolin/20 μM IBMX, and 25 μM forskolin/50 μM IBMX), respectively. The intensity of 1650 cm^−1^ was used for the evaluation of SERS readout (*n* = 3).

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
