# Peer review of "Phosphorylation-Dependent SERS Readout for Activity Assay of Protein Kinase A in Cell Extracts"

_nanomaterials, 2020, doi:10.3390/nano10030575_

Round 1
Reviewer 1 Report
Authors report a phosphorylation-dependent SERS readout strategy for protein kinase A (PKA) activity assay using spermine-functionalized silver nanoparticles (AgNPs). Protein kinases are the potentially important biomarkers, therefore the proposed method can be very promising. The manuscript can be accepted after minor revision:
- did Authors performed any post-processing of the SERS spectra? if yes, they should be described in manuscript.
- image analysis of the TEM pictures would give information about distribution of the size of the nanoparticles.
Reviewer 2 Report
Assaying protein kinase activities in real biological samples maybe a relevant issue in medical diagnostics. Literature should be integrated. Details should be given about reproducibility and aging effects. Results could be interesting, though the current version requires major revisions in the state of the art and in the discussion. Below you find detailed comments.
- Abstract
The last sentence is stating a too generic success.
- Introduction
Introduction misses a survey on metallic NP uptake in cells (see Chemical Reviews 2014, 114, 2, 1258-1288 Accounts of Chemical Research 2018, 51, 9, 2305-2313.) and known structural studies on protein-nanoparticles interaction (Nature Nanotechnology, 2015, 10, 285)
This sentence in the Introduction “SERS signal of dyes on peptides as the readout, rather than commonly labeled nanoparticles [27], could effectively exclude other experimental factors influence and avoid false positive signals” does not sound appropriate, not self-contained. Possibly, something like ”Herein we show how….”
- Results and Discussion
10 mW is a relatively high power at the sample. Did the authors performed some statistical study on the laser power (Biophys. J. 2003; 84: 3968-3981)?
More details should be provided about the time passed from SERS measurements and AgNP preparation.
More details on the particle aggregation occurring in the assay buffer should be added. Is it monodisperse, did authors observe any NP precipitation, is it time-dependent?
- 4, line 168, please include molar excess of ATP
How did authors check the effect on cell vitality with and without AgNP (Chemical Reviews 2011, 111, 5, 3407-3432 or Accounts of Chemical Research 2008, 41, 12, 1721-1730) and how AgNP affects cells features (Accounts of Chemical Research 2019, 52, 6, 1519-1530)?
How did authors check the effect of stability of AgNP (see Chemical Reviews 2015, 115, 5, 2109-2135) in Hela cells ? what about aggregation/aging effects? How long data are reproducible? Data as a function of time should be reported, since this is an issue in SERS analysis.
- Minor issues
P 4, line 140, “In principle” maybe means “in the absence of phosphorylation”?
- 5, line 171 “These…” something is missing
Round 2
Reviewer 2 Report
Revised version is now acceptable for pubblication